# Phylotypic Profiling, Distribution of Pathogenicity Island Markers, and Antimicrobial Susceptibility of *Escherichia coli* Isolated from Retail Chicken Meat and Humans

**DOI:** 10.3390/antibiotics11091197

**Published:** 2022-09-04

**Authors:** Asmaa Sadat, Hazem Ramadan, Mohamed A. Elkady, Amal Mahmoud Hammad, Mohamed M. Soliman, Salama M. Aboelenin, Helal F. Al-Harthi, Amira Abugomaa, Mohamed Elbadawy, Amal Awad

**Affiliations:** 1Department of Bacteriology, Mycology, and Immunology, Faculty of Veterinary Medicine, Mansoura University, Mansoura 35516, Egypt; 2Hygiene and Zoonoses Department, Faculty of Veterinary Medicine, Mansoura University, Mansoura 35516, Egypt; 3Mansoura Veterinary Laboratory Branch, Microbiology Research Department, Animal Health Research Institute, Kafrelsheikh 33516, Egypt; 4Biochemistry Department, Faculty of Medicine Damietta, Al-Azhar University, Cairo 11651, Egypt; 5Clinical Laboratory Sciences Department, Turabah University College, Taif University, Taif 21995, Saudi Arabia; 6Biology Department, Turabah University College, Taif University, Al Hawiyah 21995, Saudi Arabia; 7Faculty of Veterinary Medicine, Mansoura University, Mansoura 35516, Egypt; 8Department of Pharmacology, Faculty of Veterinary Medicine, Benha University, Moshtohor, Toukh 13736, Egypt

**Keywords:** *E. coli*, phylogenetic diversity, PAI markers, antimicrobial susceptibility, resistance genes

## Abstract

*Escherichia coli* (*E.coli)* found in retail chicken meat could be causing a wide range of infections in humans and constitute a potential risk. This study aimed to evaluate 60 *E. coli* isolates from retail chicken meat (*n* = 34) and human urinary tract infections (UTIs, *n* = 26) for phylogenetic diversity, presence of pathogenicity island (PAI) markers, antimicrobial susceptibility phenotypes, and antimicrobial resistance genes, and to evaluate their biofilm formation capacity. In that context, confirmed *E.coli* isolates were subjected to phylogrouping analysis using triplex PCR, antimicrobial susceptibility testing using the Kirby–Bauer disc diffusion method; PAI distribution was investigated by using two multiplex PCRs. Most of the chicken isolates (22/34, 64.7%) were identified as commensal *E. coli* (A and B1), while 12 isolates (35.3%) were classified as pathogenic virulent *E. coli* (B2 and D). Similarly, the commensal group dominated in human isolates. Overall, 23 PAIs were detected in the chicken isolates; among them, 39.1% (9/23) were assigned to group B1, 34.8% (8/23) to group A, 4.34% (1/23) to group B2, and 21.7% (5/23) to group D. However, 25 PAIs were identified from the human isolates. PAI IV536 was the most prevalent (55.9%, 69.2%) PAI detected in both sources. In total, 37 (61.7%) isolates of the chicken and human isolates were biofilm producers. Noticeably, 100% of *E. coli* isolates were resistant to penicillin and rifamycin. Markedly, all *E. coli* isolates displayed multiple antibiotic resistance (MAR) phenotypes, and the multiple antibiotic resistance index (MARI) among *E. coli* isolates ranged between 0.5 and 1. Several antibiotic resistance genes (ARGs) were identified by a PCR assay; the *sul*2 gene was the most prevalent (38/60, 63.3%) from both sources. Interestingly, a significant positive association (*r* = 0.31) between biofilm production and resistance to quinolones by the *qnr* gene was found by the correlation analysis. These findings were suggestive of the transmission of PAI markers and antibiotic resistance genes from poultry to humans or humans to humans through the food chain. To avoid the spread of virulent and multidrug-resistant *E. coli*, intensive surveillance of retail chicken meat markets is required.

## 1. Introduction 

The majority of *Escherichia coli* (*E. coli*) is found in the gastrointestinal tracts of both animals and humans as a commensal microbiota [1,2]. While certain *E. coli* strains can cause a wide range of diseases in humans and animals. [1,3], due to the increase in community-acquired human urinary tract infections (UTIs) caused by multiple antibiotic resistance (MAR) *E.*
*coli,* it is very important to recognize the most probable sources of urovirulent *E. coli* involved in such infections. Intestinal carriage of pathogenic and multidrug-resistant (MDR) *Escherichia coli* (*E. coli)* strains in food-producing animals and contamination of retail meat may contribute to increased incidences of infections by these strains in humans. Many previous studies have reported a close genetic relatedness between *E. coli* isolates from poultry and humans depending on phylogenetic groups, serogroups, and virulence gene profiles [4,5,6,7,8]. 

Poultry meat is a potential source of *E. coli* infection in humans that results in foodborne illnesses. *E. coli* is thought to enter a human host through the consumption of contaminated poultry meat. Pathogenic *E. coli* is found naturally in the guts of birds and is transferred through intestinal content, contaminating chicken carcasses during unsanitary slaughtering and processing activities, which poses a zoonotic and health hazard to people [9]. Human-to-human transmission through food is another way for poultry meat to become contaminated. Handlers transmit *E. coli* onto the meat during the preparation process, resulting in meat contamination.

Generally, *E. coli* strains are classified into four phylogenetic groups: A, B1, B2, and D [10]. According to ECOR phylogenetic grouping and virulence characteristics, both commensal and extraintestinal pathogenic *E. coli* typically differ [11]. Phylogenetic group B2 and group D are related to pathogenic extraintestinal *E. coli* isolates; however, groups A and B1 are classified as commensal *E. coli* isolates. 

Pathogenic *E. coli* strains have virulence factors [12] clustered on pathogenicity islands (PAIs) that allow them to cause infection [13,14,15]. These PAIs are a sequence of DNA <10 kb in size [16,17]. They were first identified in UPEC strain 536 [16]. PAIs and their virulence genes are disseminated between bacterial populations through horizontal transfer [15]. *E. coli* strains (*E. coli* 536, *E. coli* J96, *E. coli* CFT073) encode many PAIs; for example, PAIs I to IV from strain 536 encode fimbriae (P fimbriae, P-related fimbriae, S fimbriae, α-hemolysin, iron siderophore system, yersiniabactin siderophore system, CS12 fimbriae, and F17-like fimbrial adhesion). Furthermore, PAIs I_J96_ and II_J96_ include Prs-fimbriae, cytotoxic necrotizing factor 1, and α-hemolysin. PAIs I_CFT073_ and II_CFT073_ include aerobactin, P fimbriae, iron-regulated genes, and α-hemolysin [18]. Pathogenic ExPEC strains usually possess PAIs and belong to phylogenetic groups B2 and D [19].

During the past two decades, the development of antimicrobial resistance in Enterobacteriaceae has increased worldwide. Since *E. coli* is present as a gut commensal in humans and animals, it has become one of the microorganisms that are commonly resistant to antimicrobials. Antibiotic resistance is a rapidly growing problem due to its ability to mutate, acquire, and transmit plasmids and other mobile genetic elements that encode resistance genes [20]. In humans, ExPEC are responsible for millions of episodes of urinary tract infections (UTIs) in the United States, about 36,000 deaths from sepsis, and billions of dollars in increased healthcare costs annually [21]. Emerging antimicrobial resistance results in complications of the infections and treatment. In the poultry sector, the indiscriminate use of antimicrobials given to animals in food at subtherapeutic doses for growth promotion and prophylaxis imposes selective pressure on the organism that leads to the development of antimicrobial resistance. The emergence of antimicrobial resistance among *E. coli* of animal origin constitutes serious public health implications. Various studies reported that animals were the source of antimicrobial-resistant *E. coli* in humans [22,23,24]. Additionally, the ability of *E. coli* to produce biofilms is another source of antimicrobial resistance [25]. Biofilms can stick to the surface of numerous objects for long periods, allowing resistance genes to be transferred, which is facilitated by a protective matrix that potentiates this effect [26].

In Egypt, there is a scarcity of information on the phylogenetic grouping, virulence, resistant phenotypes, and distribution of pathogenicity island markers shared between chickens and humans. Even though PAI distributions have been investigated extensively in *E.coli* isolated from human UTI infections, little information is available regarding their characterization from poultry meat. Thus, the current study aimed to evaluate *E. coli* strains for phylogenetic profiles, antimicrobial resistance, presence of PAI markers, and biofilm formation from chicken and humans, as well as to analyze the link among the existence of the PAIs and the phylogenetic groups from both sources. 

## 2. Results

### 2.1. Prevalence of Phylogenetic Groups and PAIs among the Examined E. coli Isolates

Out of the total examined human and retail chicken meat samples (*n* = 172), 60 *E. coli* isolates were recovered with an overall prevalence of 34.9% (60/172). The recovery rates of *E. coli* from the human and chicken samples were 38.8% (26/67) and 32.4% (34/105), respectively. A PCR technique targeting a species-specific primer was used to confirm all *E. coli* isolates at the species level (Appendix A). The distribution of the four phylogroups among the chicken and human isolates was as follows: B1 (13 (38.3%) and 10 (38.5%)), A (9 (26.5%) and 4 (15.4%)), B2 (7 (20.6%) and 7 (26.9%)), and D (5 (14.7%) and 5 (19.2%)). The findings also revealed that commensal phylogroups (A and B1) were found in higher frequencies in both the chicken (22/34, 64.7%) and human (14/26, 53.8%) isolates compared to the pathogenic phylogroups (B2 and D; Table 1, Appendix A). PCR screening of pathogenic island genetic markers displayed the existence of 23 and 25 PAIs from the chicken and human isolates, respectively (Table 2). Of the 23 PAIs identified in the chicken isolates, PAI IV_536_ was the most prevalent PAI (19, 55.9%) followed by PAI II_CFT073_ (4, 11.8%), while PAI III_536_, PAI J_196_, PAI I_CFT073_, PAI II_536_, PAI I_536_, and PAI 11J_196_ were not identified. Among the human isolates, PAI IV_536_, PAI II_CFT073_, and PAI J_196_ were detected at a prevalence of 69.2% (18/26), 19.2% (5/26), and 7.7% (2/26), respectively. PAI III_536_, PAI II_536_, PAI I_CFT073,_ PAI I_536,_ and PAI IIJ_196_ were not identified (Figure 1 and Figure 2). 

### 2.2. Distribution of PAIs among the Examined E. coli Isolates in Relation to Phylogenetic Group

The distribution of PAIs in association with *E. coli* phylogroups was determined (Table 3). Higher frequencies of PAIs in the chicken isolates were detected in phylogroup B1 (9/23; 39.1%) followed by A (8/23, 34.8%), D (5/23, 21.7%), and B2 (1/23, 4.34%). In the human isolates, the distribution of PAIs in phylogenetic groups was as follows: 40% (10/25) of PAIs in phylogroup B1, 28% (7/25) in B2, 20% (5/25) in A, and 12% (3/25) in D. More than half of the identified PAIs from chicken (17/23, 73.9%) and human (15/25, 60%) isolates were assigned to the commensal phylogroups A and B1. Of the eight PAIs screened in this study, PAI IV_536_ was identified in all phylogenetic groups in isolates from both sources, PAI II_CFT073_ was characterized in three phylogenetic groups from both sources isolates, and PAI J_196_ was assigned to two phylogenetic groups among the human isolates and was not identified in the chicken isolates (Table 3). 

### 2.3. Antimicrobial Resistance Phenotypes, Genotypes, and in Vitro Biofilm Production of the Examined E. coli Isolates

The antimicrobial resistance phenotypes for the chicken and human *E. coli* isolates were determined. In both sources, 100% of *E. coli* isolates were resistant to penicillin and higher frequencies of antimicrobial resistance also were found to rifamycin (97.1% and 100%), cefuroxime (88.2% and 96.2%), cefoperazone (85.3% and 80.8%), ciprofloxacin (85.3% and 84.6%), nalidixic acid (88.2% and 92.3%), neomycin (82.4% and 92.3%), trimethoprim–sulfamethoxazole (88.2% and 76.9%), and streptomycin (73.5% and 76.9%) in the chicken and human isolates, respectively. Antimicrobial resistance to certain antimicrobials varied according to the source of the isolates. For instance, amikacin resistance was observed in 23.5% and 100% of the chicken and human isolates, respectively (Table 4). Significantly, all *E. coli* isolates from the human and chicken sources exhibited multiple antibiotic resistance. Upon calculation of MAR index values, which range from 0 to 1, all antibiotypes identified in this study had a MAR value >0.2 (Table 5). Interestingly, four isolates from chicken (*n* = 1) and human (*n* = 3) were pan-resistant to all antimicrobials used in this study. In the human and chicken isolates, 17 and 23 antibiotypes were identified, respectively. The most prevalent antibiotypes in the chicken isolates were (C, P, S, DA, STX, RA, N, NA, CXM, CEP, CIP) and (C, P, S, DA, STX, RA, N, NA, CXM, CEP, AZM, CIP), which represented approximately 30% of isolates. In human isolates, the (P, S, DA, STX, RA, N, AK, NA, CXM, CEP, AZM, CIP) antibiotype was the predominant one (5/26, 19.2%) (Table 5). 

When screening *E. coli* isolates for the presence of five antibiotic resistance genes, *sul2, bla*_TEM,_
*aphA1*, *dfrA1,* and *qnrA* were detected in 63.3%*,* 36.7%, 35%, 18.3%, and 20% respectively. Noticeably, the *sul*2 gene was found to be the most frequent (38/60, 63.3%) among *E.coli* isolates, with 85.3% (29/34) and 34.6% (9/26) from the chicken and human isolates, respectively. When compared to the human *E. coli* isolates, the presence of most of resistance genes was higher in the chicken isolates (Table 6, Appendix A). 

For both the human and chicken meat isolates, the dynamic mechanism of biofilm development was examined. In total, 37 (61.7%) isolates were found to be positive, including 23 chicken isolates and 14 human isolates. Among them, 7 (11.7%) were weak biofilm producers, 15 (25%) were moderate biofilm producers, and 15 (25%) were strong biofilm producers (Table 6). Interestingly, the biofilm-producer isolates carried most of the PAI markers compared to the non-biofilm-producer isolates. Out of 48 PAI markers identified in this study from both chicken (23) and human (25), 31 PAI markers were carried by the biofilm-producer isolates (Figure 3). Moreover, in the chicken isolates, most biofilm-producer isolates (16/23) were assigned to nonpathogenic groups A and B1, while in the human isolates, most biofilm-producer isolates were assigned to groups B1 and B2 (13/14). Noticeably, in the human isolates, all isolates of group D were non-biofilm producers (Figure 4). Concerning the association of antimicrobial resistance with biofilm production, the correlation analysis findings only revealed the existence of a significant positive association (*r* = 0.31) between biofilm production and resistance to quinolones by the *qnr* gene (Appendix A). 

Hierarchal clustering of the chicken and human isolates was performed based on the presence/absence of PAIs among isolates. Chicken and human *E. coli* were grouped into three major clusters (A, B, and C); isolates in each cluster had identical profiles of PAIs (Figure 5). Both clusters A and B were composed of isolates from both sources, while cluster C contained two isolates sourced from human urine. Within each cluster, the distribution of antimicrobial resistances, phylogenetic groups, and biofilm production were not confined to a certain isolate source.

## 3. Discussion

Retail meats and poultry serve as an important vehicle for transmitting antimicrobial-resistant and/or pathogenic bacteria to consumers. Pathogenic bacteria with virulence and antimicrobial-resistance characteristics can contaminate poultry meat in large quantities during slaughtering procedures [27]. Food handlers may also transmit pathogenic *E. coli* from one person to another through food in processing plants. It was assumed that *E. coli* isolated from poultry and human were completely different; however, recent investigations of both intestinal and extraintestinal pathogenic *E. coli* isolated from both sources revealed common genetic features [4,5,6,7,8,28,29]. Serogrouping, phylogenetic grouping, virulence genotyping, antimicrobial sensitivity testing, and pathogenicity testing were used frequently to investigate this connection [28,30,31,32,33,34]. In this study, we collected meat and urine samples from chickens and humans, respectively, to study the phylogrouping, distribution of pathogenicity island markers, antimicrobial-resistant genotypes, and phenotypes shared between chickens and humans. Out of 105 fresh chicken meat and 67 urine samples from human urinary tract infections, 34 (32.38%) and 26 (38.8%) *E.coli* isolates were recovered, respectively. All the recovered *E. coli* isolates were subjected to PCR-based phylotyping that provided information about virulence based on their grouping into pathogenic and commensal groups. The *E. coli* isolates were grouped into four primary phylogenetic groups; namely, A, B1, B2, and D. The majority of pathogenic virulent *E. coli* isolates belonged to phylogenetic groups B2 and D [10]; the virulence genes were mostly associated with phylogroups B2 and D [35,36]. In this study, the presence of phylogroups B1 and A in approximately 38.3% and 26.5%, respectively, of chicken isolates, which was predominantly associated with commensal *E. coli*, could be attributed to the origin of these strains as commensals that acquired virulence-related genes. Similarly, commensal group A dominated among avian pathogenic *E. coli* strains in previous studies conducted in the USA, Germany, and Brazil [5,37,38]. On the other hand, pathogenic groups B2 and D were detected in 20.6%% and 14.7%, respectively. Both groups B2 and D are of public health concern because they are thought to be more pathogenic and virulent. As a result, pathogenic phylogenetic groups of chicken isolates could constitute a public health risk through direct contact or contaminated chicken products [38,39].

Regarding the human isolates, the phylotype distribution was as follows: B1 (10/26, 38.5%), B2 (7/26, 26.9%), D (5/26, 19.2%) and A (4/26, 15.4%), with predominance of phylogenetic group B1. This finding agreed with previously reported studies that showed higher frequencies of phylogroup B1 amongst *E. coli* isolates [40,41] and the infrequency of group A [42,43]. However, the phylogenetic group B2 was reported as the most prevalent group in another study of UTIs [44,45,46]. Many factors influence *E. coli*’s phylogenetic distribution, including geographical dispersion, host genetic factors, dietary factors, and medicine used [40,44,47].

The presence of different virulence genes on plasmids or chromosomal areas, known as pathogenicity islands (PAIs), is usually connected with *E. coli’s* potential to induce infection [48]. Many studies have focused on the characterization of PAIs from UPEC, but there is still a gap in the knowledge concerning their characterization in non-UPEC isolates. [49,50]. Therefore, in this study, we studied the distribution of PAI markers among both chicken and human isolates. Overall, 23 PAI markers were identified from chicken isolates, and PAI IV_536_ was the most prevalent PAI identified. The high frequency of PAI IV_536_ contributed to its stability because it was the first to be acquired and fixed on the chromosome, while the other PAIs were unstable and easy to lose [51]. 

Most of the human isolates (*n* = 20) harbored PAIs markers; likewise, a lower prevalence was reported in [14] and [52], while a higher occurrence of PAIs from *E. coli* isolated from UTIs was reported [12,53,54]. In this study, PAI IV_536_ (69.2%), PAI II_CFT073_ (19.2%), and PAI J_196_ (7.7%) were identified in human isolates. In detail, PAI IV_536_ was detected in 13 isolates, PAI J_196_ was detected in 2 isolates, and both PAI IV_536_ and _PAI_ II_CFT073_ were identified in 5 isolates. In total, 20 isolates out of 26 human isolates (76.9%) harbored PAIs. The most frequent combination of PAI markers among chicken and human isolates was PAI IV_536_+PAI II_CFT073._ These findings were in agreement with previous reports on humans in Iran [53], China [55], Sweden [36], and Spain [56]. A similar distribution of PAIs with a predominance of PAI IV_536_ was reported in many studies [12,55,56,57,58,59]. 

The identification of PAI markers on a regular basis is thought to be a beneficial tool for detecting pathogenic *E. coli* pathotypes and, as a result, introducing new therapeutic targets [16]. The number of PAIs found in this investigation varied among phylogroups but was considerably high in groups A (34.8%) and B1 (39.1%) in the chicken and human isolates than in the other groups. However, prior research on *E. coli* isolates from UTIs indicated that the group B1, A, and D UPEC isolates had fewer PAI markers than group B2. [12,56]. 

PAIs are genomic parts that contain virulence genes responsible for the pathogenicity of bacteria. Although the majority of PAIs are not mobile, additional mobile genetic elements such as conjugative plasmids, bacteriophages, or integrative and conjugative elements (ICEs) can help them spread among bacterial species [18,60,61]. Based on the presence/absence of PAIs, we discovered that *E. coli* isolates from humans and chickens were clustered together. Most importantly, the existence of isolates that exhibited identical PAI patterns and that belonged to the commensal phylogroup B1 could have been the reason for the possible acquisition of PAIs by commensal *E. coli* [55].

Multiple antibiotic resistance in *E. coli* has emerged as a serious problem in both human and veterinary medicine. Most *E. coli* isolates were found to have a high frequency of resistance in this study. They were extremely resistant to β-lactams, sulfonamides, aminoglycosides, quinolones, lincosamide, fluoroquinolones, and rifampin regardless of the source of *E. coli* isolates, while slightly lower frequencies of resistance were detected against macrolide (azithromycin), amikacin, and chloramphenicol. These marked antibiotic resistances may contribute to the dissemination of resistance plasmids, which are involved in the spreading of resistance genes between veterinary and human healthcare [62]; specifically, those coding for extended-spectrum lactamases, 16S rRNA methylases, and plasmid-mediated quinolone resistance (PMQR), which confer resistance to broad-spectrum cephalosporins, aminoglycosides, and fluoroquinolones, respectively. Noticeably, MAR was found in all chicken and human isolates against three or more antimicrobial drugs. Furthermore, the MAR of retail chicken meat was greater than 0.6, indicating a high risk of human infection. Long-term exposure and/or indiscriminate use of various antimicrobials in poultry farms for preventive and therapeutic purposes, as well as in human urinary tract infection, may be responsible for these high levels of antimicrobial resistance, which pose major health risks to both animals and humans. Other theories suggest that the transfer of resistance genes via plasmids and other mobile genetic elements is a key process in the development of multiple antibiotic resistance in *E. coli*. Increased frequencies of *sul*2, *bla*_TEM_, and *apha*1 genes were found in the investigated isolates, which corresponded to higher resistance phenotypes to the respective antimicrobials (sulfamethoxazole trimethoprim, penicillin, and streptomycin). The pan-resistant strains were mostly assigned to commensal group B1 and were biofilm producers, while one human isolate was assigned to the pathogenic *E.coli* (group D). Regarding the distribution of PAIs, two strains harbored PAI IV536, while the others tested negative for PAIs. This finding highlighted the emergence of acquiring AMR between *E. coli* strains regardless of the origin or the phylogeny. 

The main complication of *E. coli* infection is the formation of a biofilm, which is a significant virulence factor. In slaughterhouses and food-processing plants, biofilm growth can result in financial losses due to equipment and facility failures, as well as food spoilage. [63]. Furthermore, biofilm-forming bacteria pose a public health danger because they can adhere to a variety of surfaces and biofilm structures can be broken, resulting in the release of hazardous microbes and product contamination [64]. In humans, biofilms provide bacteria with a survival strategy by allowing them to make the best use of available nutrients while preventing access to antimicrobial drugs, antibodies, and white blood cells [65]. They have also been found to contain a high number of antibiotic-inactivating enzymes such as beta-lactamases, resulting in an antimicrobial resistance island [66]. Controlling infections induced by *E. coli* biofilms is difficult due to the extracellular matrix and the observed increased resistance to common antibiotics. In this study, 61.7% (37/60) of *E.coli* isolates were biofilm-formers, including 23 (67.4%) isolates from chickens and 14 (58.8%) isolates from humans. These findings were similar to those reported in strains from birds [67] and humans [68]. This highlighted the importance of investigating alternative therapeutics such as antiadhesion agents, phytochemicals, and nanomaterials for effective drug delivery in humans to limit biofilm formation, as well as the need to improve cleaning and disinfection measures at slaughterhouses.

## 4. Materials and Methods

### 4.1. Sample Collection and Conventional Isolation of E. coli

In this study, a total of 172 samples from humans (*n* = 67) and retail chicken meat (*n* = 105) were collected between January and July 2019 and examined for the presence of *E. coli*. An informed consent for the current study was obtained from patients before urine sampling. Midstream urine samples were taken from patients who were clinically diagnosed with urinary tract infections (UTIs) and admitted to the Specialized Medical Hospital in Mansoura, Egypt. The authors were not involved in the sampling process for urine from the patients. Retail chicken meats were randomly collected weekly from retail chicken shops in Mansoura, Egypt. Samples were transported under complete aseptic conditions in an icebox to the laboratory of the Department of Bacteriology, Mycology, and Immunology, Faculty of Veterinary Medicine, Mansoura University, Mansoura, and subjected to conventional bacteriological analysis. 

First, 1 mL of urine sample was enriched in 9 mL of tryptic soy broth (TSB; Oxoid, UK) and incubated at 37 °C for 18 h. Retail chicken meat (25 g) was homogenized in TSB (225 mL) and ground in a stomacher, and the homogenate was incubated overnight at 37 °C. A loopful of the enriched broth from the retail chicken meat and human urine samples was plated onto the surface of MacConkey agar plates (HiMedia, Mumbai, India) and incubated at 37 °C for 18–24 h. Lactose fermenter colonies (pink colonies) were picked up from MacConkey agar and subcultured on Eosin Methylene Blue (EMB) agar (HiMedia, Mumbai, India). Presumptive *E. coli* colonies with the characteristic green color and metallic sheen were subjected to the standard morphological and biochemical tests [69].

### 4.2. Molecular Characterization of E. coli

In 200 μL of deionized water, three to five bacterial colonies were picked up, heated for 15 min, and then centrifuged for three min at 10,000 rpm. After that, the supernatants were transferred to sterile, clean Eppendorf tubes and kept until used as a DNA template [70]. To confirm the genus identity of presumptive *E. coli* isolates, a uniplex PCR assay was conducted using a genus-specific primer (Table 7) that targeted variable regions within the 16S small subunit rRNA gene [71]. 

### 4.3. PCR-based Phylotyping 

*Escherichia coli* isolates were subjected to a phylogrouping analysis using triplex PCR as proposed by Clermont et al. [10] with three primer pairs: chuA.1–chuA.2, yjaA.1–yjaA.2, and TspE4C2.1–TspE4C2.2 (Table 7). Strains were assigned to four main phylogenetic groups (A, B1, B2, or D) according to the amplification of the three primer pairs [30]. A reaction mixture of 23 μL containing 3 μL of DNA template, 10 mL of 2X PCR master mix (Thermo scientific, Waltham, MA, USA), 4 μL of deionized free water, and 1 μL of every one of the six primers. PCR reactions were applied using 96-well plates and an Applied Biosystem 2720 thermal cycler; the cyclic PCR conditions were as follows: initial denaturation at 94°C for 5 min; 30 cycles of 94 °C for 30 sec, 55 °C for 30 sec, and 72 °C for 30 sec; and a final extension at 72 °C for 7 min. Amplicons (5 μL) were separated in 1% agarose gel stained with ethidium bromide and visualized under a UV transilluminator and a Gel Documentation System (cleaver scientific ltd UV gel documentation system, Rugby, Warwickshire, UK). The results were interrupted according to Bonacorsi et al. [77].

### 4.4. Detection of Pathogenicity Islands Markers

Eight PAIs in *E. coli* isolates were investigated in this study using two multiplex PCRs (1 and 2) according to Sabaté et al [56]. Three PAI markers were targeted (PAI III536, PAI IV536, and PAI IICFT073) in the first reaction, yielding 200, 300, and 400 bp, respectively; the reaction mixture measured 20 μL and contained 3 μL of DNA template, 10 μL of 2X PCR master mix (Thermo scientific, Waltham, MA, USA), 6 μL of deionized water, and 1 μL of each of the six primers. The second multiplex PCR targeted five PAI markers (PAI IIJ96, PAI I536, PAI II536, PAI ICFT073, and PAI IJ96), yielding 2300, 1800, 1000, 930, and 400 bp fragments, respectively. The PCR reaction was performed in a total volume of 20 μL containing 3 μL of DNA template, 10 μL of 2X PCR master mix (Thermo scientific, Waltham, MA, USA), 2 μL of deionized water, and 1 μL of each primer. The primer sequences and expected sizes of amplicons for each PCR assay are given in Table 7. Both multiplex PCRs were performed under the following conditions: 94 °C for 5 min; 30 cycles of 94 °C for 1 min, 55 °C for 1 min, and 72 °C for 1 min; and a final extension at 72 °C for 10 min. 

### 4.5. Antimicrobial Susceptibility Testing 

The Kirby–Bauer disc diffusion method was used to test the susceptibility of the *E. coli* isolates to antimicrobial drugs [78]. A total of 13 antimicrobial discs (Oxoid, Basingstoke, UK) were selected, including penicillin (P; 10 μg), cefuroxime (CXM; 30 μg), cefoperazone (CEP; 30 μg), amikacin (AK; 30 μg), streptomycin (S; 15 μg), neomycin (N; 5 μg), azithromycin (AZM; 15 μg), nalidixic acid (NAL; 30 μg), trimethoprim–sulfamethoxazole (SXT; 25 μg), clindamycin (DA; 2 μg), ciprofloxacin (CIP; 5 μg), chloramphenicol (C; 30 μg), and rifampin (RA; 5 μg). These antibiotics are widely used in clinical and veterinary medicine. The percentages, frequencies, and profile of antibiotic-resistant *E. coli* were obtained and phenotyped for their multiple antibiotic resistance using *E. coli* ATCC 25922 as a quality control. The isolates that exhibited resistance ≥3 of the antimicrobial classes were recorded. The MAR index was derived by dividing the total number of antimicrobial resistances for each isolate by the total number of tested antimicrobials [79]; this is a crucial indicator in determining the risk source of contamination that is a possible hazard to humans [80]. The results were interpreted according to the interpretative chart in [78].

### 4.6. Molecular Characterization of Antimicrobial Resistance Genes 

*E.coli* isolates were screened using PCR for characterization of the antimicrobial resistance genes that encoded resistance to β-lactamase (bla*_TEM_*), quinolone (*qnrA*)*,* trimethoprim (*dfrA1*)*,* aminoglycoside (*aphA1*), and sulfonamide (*sul1*). For bla*_TEM_*, *qnrA,* and *dfrA1,* these were examined as previously described in [73,75,76], respectively, while *aphA1 and sul2* were investigated according to the procedure by Maynard et al. [74]. A reaction mixture of 25 μL containing 5 μL of DNA template, 12.5 μL of 2X PCR master mix (Thermo scientific, United States), 5.5 μL of deionized free water, and one μL of each primer was performed.

### 4.7. Biofilm-Formation Assay 

The capacity of the *E. coli* isolates to produce biofilms was assessed according to [81]. Briefly, *E. coli* isolates were grown individually in glass tubes containing TSB for 10 h at 28 °C. The inoculated broth was discarded hygienically and the tubes were dyed with a 1% crystal violet solution [82]. After 15 min, the stained tubes were washed with sterile distilled water. All isolates, including the uninoculated TSB used as a negative control, were examined in triplicate. The stained tubes were examined for a visible film lining their walls and bottoms.

The results for PAI marker screening, resistance phenotypes and genotypes, phylogenetic groups, and biofilm production were imported into the BioNumerics software (Applied Maths, Austin, TX, USA, version 7.6) to generate a dendrogram for clustering the examined isolates from humans and chicken meat. The dendrogram was generated based on the PAI results using the band-based Dice coefficient and the unweighted pair group method with arithmetic averages (UPGMA). A correlation analysis to determine the association of antimicrobial resistance phenotypes, resistance genotypes, and biofilm production among the examined isolates was performed as previously described [83,84] (add these references please).

## 5. Conclusions 

The findings of this study were suggestive of the transmission of PAI markers and antibiotic resistance genes from poultry to humans or humans to humans through the food chain, which could be a potential source of human disease. Thus, to avoid the spread of virulent and multidrug-resistant *E. coli*, intensive surveillance of retail chicken meat markets is required.

## Figures and Tables

**Figure 1 antibiotics-11-01197-f001:**
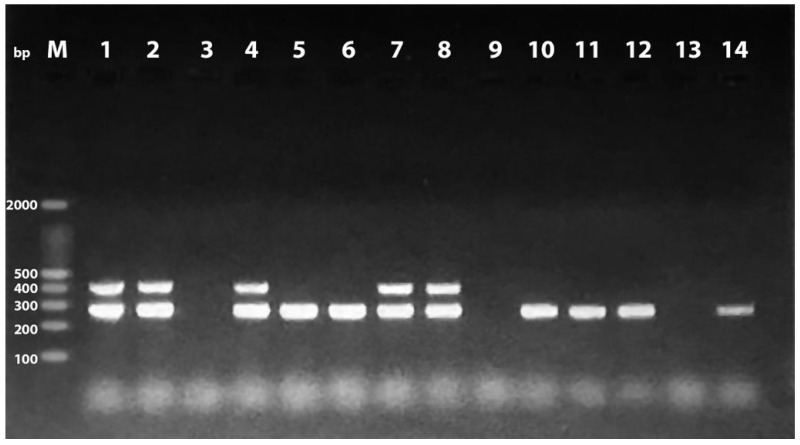
Multiplex PCR for identification of PAI IV_536_ and PAI II_CFT073_ at 300 and 400 bp. Lane M: 100 bp DNA ladder; lanes 1, 2, 4, 7, and 8: positive for both PAI IV_536_ and PAI II_CFT073_; lanes 5, 6, 10, 11, 12, and 14: positive for PAI II_CFT073_; lanes 3, 9, and 13: negative samples.

**Figure 2 antibiotics-11-01197-f002:**
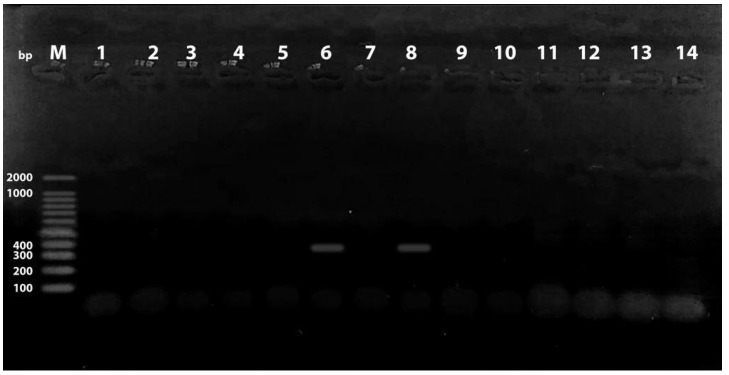
Multiplex PCR showing amplification of PAI IJ96 at 400 bp. Lane M: 100 bp DNA ladder; lanes 6 and 8: positive samples; lanes 1–5, 7, and 9–14: negative samples.

**Figure 3 antibiotics-11-01197-f003:**
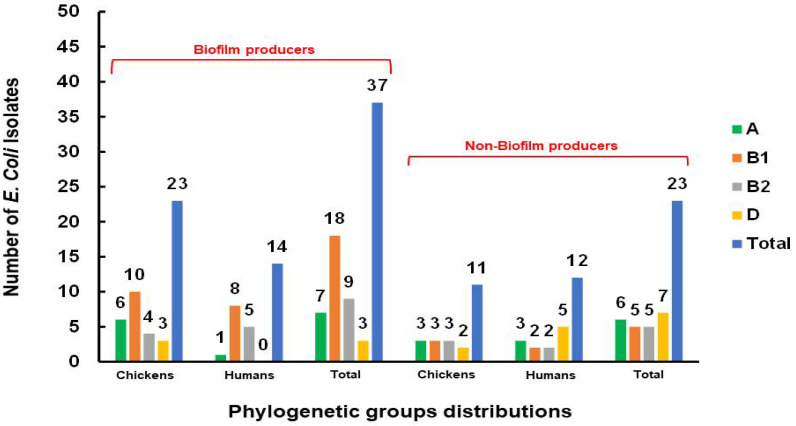
Distribution of phylogenetic groups among *E. coli* isolates in relation to biofilm production.

**Figure 4 antibiotics-11-01197-f004:**
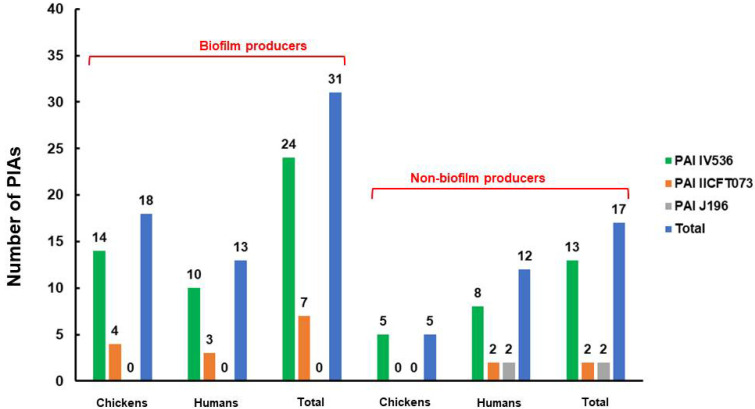
Distribution of pathogenicity islands (PAIs) among *E. coli* isolates in relation to biofilm production.

**Figure 5 antibiotics-11-01197-f005:**
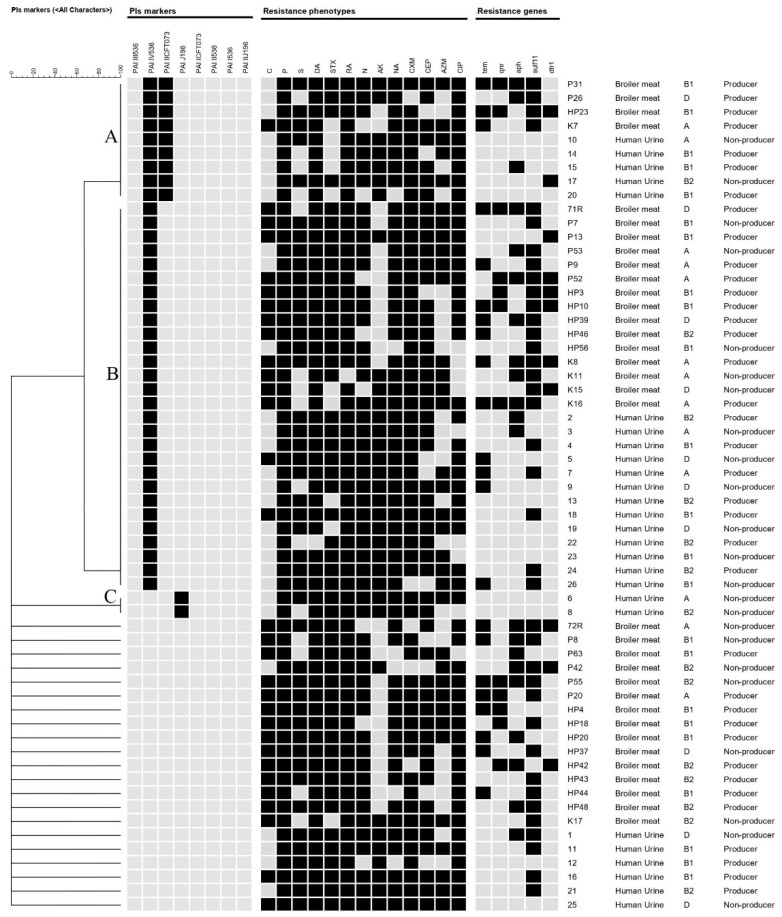
Hierarchal clustering of chicken and human isolates based on the existence of pathogenic islands (PAIs) using Dice coefficient and the unweighted pair group method (UPGMA) in BioNumerics software (Applied Maths, Austin, TX, USA, version 7.6). Black and gray squares represent positive and negative results, respectively, for PAIs, resistant phenotypes, and resistant genotypes. Phylogroups and biofilm production results were also determined.

**Table 1 antibiotics-11-01197-t001:** Frequency distribution of phylogenetic groups among the examined *E. coli* isolates from chicken and human.

Source of Isolates	Phylogenetic Group					
	Pathogenic Phylogroups			Commensal Phylogroups		
	Group B2	Group D	Total	Group A	Group B1	Total
Chicken (*n* = 34)	7 (20.6%)	5 (14.7%)	12 (35.3%)	9 (26.5%)	13 (38.3%)	22 (64.7%)
Human (*n* = 26)	7 (26.9%)	5 (19.2%)	12 (46.2)	4 (15.4%)	10 (38.5%)	14 (53.8%)

**Table 2 antibiotics-11-01197-t002:** Pathogenicity island (PAI) markers identified in *E. coli* isolates from chicken and human.

Source of Isolates	PAI III536	PAI IV536	PAI IICFT073	PAI J196	PAI ICFT073	PAI II536	PAI I536	PAI IIJ196
Chicken	-	19 (55.9%)	4 (11.8%)	-	-	-	-	-
Human	-	18 (69.2%)	5 (19.2%)	2(7.7%)	-	-	-	-

**Table 3 antibiotics-11-01197-t003:** Distribution of pathogenicity islands (PAIs) among *E. coli* isolates in relation to phylogenetic group.

Isolates	PAIs/Phylogenetic Group	PAI III536	PAI IV536	PAI IICFT073	PAI J196	PAI ICFT073	PAI II536	PAI I536	PAI IIJ196	Total
Chicken (*n* = 34)	B2	0	1	0	0	0	0	0	0	1 (4.34%)
	D	0	4	1	0	0	0	0	0	5 (21.7%)
	A	0	7	1	0	0	0	0	0	8 (34.8%)
	B1	0	7	2	0	0	0	0	0	9 (39.1%)
	Total	0	19	4	0	0	0	0	0	23 (100%)
Human (*n* = 26)	B2	0	5	1	1	0	0	0	0	7 (28%)
	D	0	3	0	0	0	0	0	0	3 (12%)
	A	0	3	1	1	0	0	0	0	5 (20%)
	B1	0	7	3	0	0	0	0	0	10 (40%)
	Total	0	18	5	2	0	0	0	0	25 (100%)

**Table 4 antibiotics-11-01197-t004:** Results of antimicrobial susceptibility testing for the examined *E. coli* isolates.

Antibiotics	Family	Disc Code	CPD	Chicken (34)		Human (26)	
				Resistant	Sensitive	Resistant	Sensitive
Penicillin	β-lactam	P	10	34 (100%)	0 (0.00%)	26 (100%)	0 (0.00%)
Cefuroxime	Cephalosporin	CXM	30	30 (88.2%)	4 (11.8%)	25 (96.2%)	1 (3.9%)
Cefoperazone		CEP	30	29 (85.3%)	5 (14.7%)	21 (80.8%)	5 (19.2%)
Amikacin	Aminoglycoside	AK	30	8 (23.5%)	26 (76.5%)	26 (100%)	0 (0.00%)
Streptomycin		S	15	25 (73.5%)	9 (26.5%)	20 (76.9%)	6 (23.1%)
Neomycin		N	5	28 (82.4%)	6 (17.6%)	24 ( 92.3%)	2 ((7.7%)
Azithromycin	Macrolide	AZM	15	20 (58.8%)	14 (41.2%)	15 (57.7%)	11 (42.3%)
Nalidixic acid	Quinolone	NAL	30	30 (88.2%)	4 (11.8%)	24 (92.3%)	2 (7.7%)
Trimethoprim–sulfamethoxazole	Sulfonamide	SXT	25	30 (88.2%)	4 (11.8%)	20 (76.9%)	6 (23.1%)
Clindamycin	Lincosamide	DA	2	34 (100%)	0 (0.00%)	25 (96.2%)	1 (3.8%)
Ciprofloxacin	Fluoroquinolone	CIP	5	29 (85.3%)	5 (14.7%)	22 (84.6%)	4 (15.4%)
Chloramphenicol	Phenicols	C	30	27 (79.4%)	7 (20.6%)	4 (15.4%)	22 (84.6%)
Rifampin	Rifamycin	Rifampin	5	33 (97.1%)	1 (2.9%)	26 (100%)	0 (0.00%)

**Table 5 antibiotics-11-01197-t005:** Different patterns of antimicrobial susceptibility detected for the *E. coli* isolates.

	Antibiotypes	Resistance Pattern	Isolates No. (%)	MAR	MAR Index
Chicken	I	C, P, DA, STX, RA, N, CXM, CIP	1 (2.9%)	8	0.62
	II	P, S, DA, STX, RA, N, CXM, CEP	1 (2.9%)	6	0.62
	III	C, P, S, DA, SXT, RA, NA, CEP, CIP	1 (2.9%)	9	0.7
	IV	C, P, DA, STX, RA, N, NA, CXM, CIP	1 (2.9%)	9	0.7
	V	C, P, DA, STX, RA, N, CXM, CEP, AZM	1 (2.9%)	8	0.7
	VI	P, S, DA, STX, RA, N, AK, AZM, CIP	1 (2.9%)	8	0.7
	VII	P, DA, STX, RA, N, AK, NA, CEP, CIP	1 (2.9%)	8	0.7
	VIII	P, S, DA, STX, RA, N, NA, CXM, CIP	1 (2.9%)	8	0.7
	IX	C, P, DA, RA, AK, NA, CXM, CEP, AZM	1 (2.9%)	8	0.7
	X	C, P, S, DA, STX, RA, N, NA, CXM, CIP	1 (2.9%)	9	0.77
	XI	C, P, S, DA, STX, RA, N, NA, CEP, CIP	1 (2.9%)	9	0.77
	XII	C, P, S, DA, STX, RA, NA, CXM, CEP, CIP	1 (2.9%)	9	0.77
	XIII	C, P, S, DA, RA, NA, CXM, CEP, AZM, CIP	1 (2.9%)	9	0.77
	XIV	C, P, DA, STX, N, AK, NA, CXM, CEP, AZM	1 (2.9%)	8	0.77
	XV	C, P, DA, STX, RA, N, NA, CXM, CEP, AZM, CIP	1 (2.9%)	10	0.84
	XVI	P, S, DA, STX, RA, N, NA, CXM, CEP, AZM, CIP	2 (5.8%)	9	0.84
	XVII	C, P, S, DA, STX, RA, NA, CXM, CEP, AZM. CIP	2 (5.8%)	10	0.84
	XVIII	C, P, S, DA, STX, RA, N, NA, CXM, CEP, CIP	5 (14.7%)	9	0.84
	XIX	C, P, S, DA, STX, RA, N, NA, CXM, CEP, AZM	1 (2.9%)	9	0.84
	XX	C, P, DA, RA, N, AK, NA, CXM, CEP, AZM, CIP	2 (5.8%)	9	0.84
	XXI	C, P, S, DA, STX, RA, N, NA, CXM, CEP, AZM, CIP	5 (17.7%)	10	0.9
	XXII	P, S, DA, STX, RA, N, AK, NA, CXM, CEP, AZM, CIP	1 (2.9%)	9	0.9
	XXIII	C, P, S, DA, STX, RA, N, AK, NA, CXM, CEP, AZM, CIP	1(2.9%)	10	1
Human	I	P, DA, RA, AK, CXM, CEP, CIP	1 (3.8%)	6	0.54
	II	P, S, DA, STX, RA, AK, CXM, CIP	1 (3.8%)	7	0.62
	III	P, STX, RA, N, AK, NA, CXM, CEP	1 (3.8%)	7	0.62
	IV	P, DA, SXT, RA, N, AK, NA, CXM, CEP	1 (3.8%)	8	0.7
	V	P, DA, RA, N, AK, NA, CXM, AZM, CIP	1 (3.8%)	8	0.7
	VI	P, DA, RA, N, AK, NA, CXM, CEP, CIP	1 (3.8%)	7	0.7
	VII	P, S, DA, STX, RA, N, AK, NA, CXM, CEP	1 (3.8%)	8	0.77
	VIII	P, S, DA, RA, N, AK, NA, CXM, CEP, CIP	1 (3.8%)	7	0.77
	IX	P, S, DA, STX, RA, N, AK, NA, AZM, CIP	1 (3.8%)	8	0.77
	X	P, S, DA, SXT, RA, N, AK, NA, CXM, CEP, CIP	3 (11.5%)	8	0.85
	XI	C, P, S, DA, SXT, RA, N, AK, NA, CXM, CIP	1 (3.8%)	9	0.85
	XII	P, S, DA, STX, RA, N, AK, NA, CXM, AZM, CIP	1 (3.8%)	10	0.85
	XIII	P, DA, STX, RA, N, AK, NA, CXM, CEP, AZM, CIP	1 (3.8%)	9	0.85
	XIV	P, S, DA, RA, N, AK, NA, CXM, CEP, AZM, CIP	2 (7.7%)	8	0.85
	XV	P, S, DA, STX, RA, N, AK, NA, CXM, CEP, AZM	1 (3.8%)	8	0.85
	XVI	P, S, DA, STX, RA, N, AK, NA, CXM, CEP, AZM, CIP	5 (19.2%)	9	0.92
	XVII	C, P, S, DA, STX, RA, N, AK, NA, CXM, CEP, AZM, CIP	3 (11.5%)	10	1

**Table 6 antibiotics-11-01197-t006:** Prevalence of antimicrobial resistance genes and biofilm production detected in *E. coli* isolates.

Point of Differences	Items Tested	Chicken Isolates (34)	Human Isolates (26)	Total(60)
Resistant genes	*bla* _TEM_	18 (52.9%)	4 (15.4%)	22 (36.7%)
	*qnrA*	12 (35.3%)	0 (0.00%)	12 (20%)
	*aphA1*	17 (50%)	4 (15.4%)	21 (35%)
	*sul2*	29 (85.3%)	9 (34.6%)	38 (63.3%)
	*dfrA1*	10 (29.4%)	1 (3.8%)	11 (18.3%)
Biofilm production	Negative	11 (32.4%)	12 (46.2%)	23 (38.3%)
	Weak positive	4 (11.8%)	3 (11.5%)	7 (11.7%)
	Moderate	11 (32.4%)	4 (15.4%)	15 (25%)
	Strong positive	8 (23.5%)	7 (26.9%)	15 (25%)
	Overall positive	23 (67.6%)	14 (53.8%)	37 (61.7%)

**Table 7 antibiotics-11-01197-t007:** List of oligonucleotide primers used in the present study.

Gene	Primer Name	Primer Sequences (5’ to 3’)	(bp)	Reference
16S rRNA	16S-F	F-GCGGACGGGTGAGTAATGT	200	[71]
	16S-R	R-TCATCCTCTCAGACCAGCTA		
Pathogenicity Island (PAI) Genes				
PAI I_536_	I.9	F-TAATGCCGGAGATTCATTGTC	1800	[56]
	1.10	R-AGGATTTGTCTCAGGGCTTT		
PAI II_536_	orf1 up	F-CCATGTCCAAAGCTCGAGC	1000	
	orf1 down	R-CTACGTCAGGCTGGCTTTG		
PAI III_536_	sfaAI.1	F-CGGGCATGCATCAATTATCTTTG	200	[3]
	sfaAI.2	R-TGTGTAGATGCAGTCACTCCG		
PAI IV_536_	IRP2 FP	F-AAGGATTCGCTGTTACCGGAC	300	[3]
	IRP2 RP	R-TCGTCGGGCAGCGTTTCTTCT		
PAI II_CFT073_	cft073.2Ent1	F-ATGGATGTTGTATCGCGCP	400	[56]
	cft073.2Ent2	R-ACGAGCATGTGGATCTGC		
PAI J_196_	papGIf	F-TCGTGCTCAGGTCCGGAATTT	400	[13]
	papGIR	R-TGGCATCCCACATTATCG		
PAI IIJ_196_	Hlyd	F-GGATCCATGAAAACATGGTTAATGGG	2300	
	Cnf	R-GATATTTTTGTTGCCATTGGTTACC		
PAI I_CFT073_	RPAi	F-GGACATCCTGTTACAGCACGCA	930	
	RPAf	R-TCGCCACCAATCACAGCCGAAC		
Phylogenetic Island Genes				
*Chu*A	ChuA-F	F-GACGAACCAACGGTCAGGAT	279	[72]
	ChuA-R	R-TGCCGCCAGTACCAAAGACA		
*Yja*A	YjaA-F	F-TGAAGTGTCAGGAGACGCTG	211	
	YjaA-R	R-ATGGAGAATGCGTTCCTCAAC		
TspE4C2	TspE4C2-F	F-GAGTAATGTCGGGGCATTCA	152	
	TspE4C2-R	R-CGCGCCAACAAAGTATTACG		
Antimicrobial Resistance Genes				
*bla* _TEM_	Tem-F	ATTCTTGAAGACGAAAGG	1150	[73]
	Tem-R	ACGCTCAGTGGAACGAAAAC		
*sul*2	Sul2-F	CGGCATCGTCAACATAACC	722	[74]
	Sul2-R	GTGTGCGGATGAAGTCAG		
*dfr*A1	dfrA1-F	GGAGTGCCAAAGGTGAACAGC	367	[75]
	dfrA1-R	GAGGCGAAGTCTTGGGTAAAAAC		
*qnr*A	QnrA-F	ATTTCTCACGCCAGGATTTG	514	[76]
	QnrA-R	GATCGGCAAAGGTTAGGTCA		
*aph*A1	aphA1-F	ATGGGCTCGCGATAATGTC	600	[74]

## Data Availability

The datasets generated during and/or analyzed during the current study are available from the corresponding author on reasonable request.

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
