# Peer review of "Phylotypic Profiling, Distribution of Pathogenicity Island Markers, and Antimicrobial Susceptibility of *Escherichia coli* Isolated from Retail Chicken Meat and Humans"

_antibiotics, 2022, doi:10.3390/antibiotics11091197_

Round 1

Reviewer 1 Report

The manuscript entitled “Phylotypic profiling, distribution of pathogenicity island markers, and antimicrobial susceptibility of Escherichia coli isolated from retail chicken’s meat and humans” is well written. This manuscript reports the results of antimicrobial susceptibility testing for the examined E. coli isolates from retail chicken meat and human urinary tract infections. The authors’ findings are suggestive of the transmission of islands markers and antibiotic resistance genes from poultry to humans or human to human through the food chain which could be a potential source of human disease. The conclusion is in agreement with the results reported. However, I do not see many novelty in the manuscript.

Minor comments:

1) In Figures 1,2, S1-S3, the interpretations of samples, bands, etc., should be provided. Figure legends should be able to stand alone. Readers should be able to understand them without referring to the text.

2) Lines 212-214:"Chicken and human E. coli were grouped into three major clusters (A, B, C), where isolates in each cluster had identical profiles of PAIs (Fig. 5)". In Fig. 5 appropriate symbols (A, B, C) should be added.

3) The section “3. Discussion” can be significantly improved by considering these and other recent works:

Gultekin, E. O., Ulger, S. T., & Delialioğlu, N. (2022). Distribution of Pathogenicity Island Markers and Virulence Factors Genes of Extraintestinal Pathogenic Escherichia coli Isolates. Jundishapur Journal of Microbiology, 15(4).

Kuznetsova, M. V., Maslennikova, I. L., Pospelova, J. S., Bertok, D. Ž., & Erjavec, M. S. (2022). Differences in recipient ability of uropathogenic Escherichia coli strains in relation with their pathogenic potential. Infection, Genetics and Evolution, 97, 105160.

El-Shaer, S., Abdel-Rhman, S. H., Barwa, R., & Hassan, R. (2021). Genetic characterization of extended-spectrum β-Lactamase-and carbapenemase-producing Escherichia coli isolated from Egyptian hospitals and environments. PloS one, 16(7), e0255219.

El-Mahdy, R., Mahmoud, R., & Shrief, R. (2021). Characterization of E. coli phylogroups causing catheter-associated urinary tract infection. Infection and Drug Resistance, 14, 3183.

Author Response

The manuscript entitled “Phylotypic profiling, distribution of pathogenicity island markers, and antimicrobial susceptibility of Escherichia coli isolated from retail chicken’s meat and humans” is well written. This manuscript reports the results of antimicrobial susceptibility testing for the examined E. coli isolates from retail chicken meat and human urinary tract infections. The authors’ findings are suggestive of the transmission of islands markers and antibiotic resistance genes from poultry to humans or human to human through the food chain which could be a potential source of human disease. The conclusion is in agreement with the results reported. However, I do not see many novelty in the manuscript.

Frist of all, the authors would like to thank the reviewer for his valuable comments which undoubtedly will improve our manuscript. All comments were considered and addressed as suggested. In the introduction section, we stated that the distribution of PAIs from human E.coli isolates were frequently studied before, but in our study we study their distribution from human as well as poultry meat isolates to highlight the potential zoonotic hazard from the contamination of poultry meat by these virulence strains and its effect on public health. In addition, there are many relationship were investigated in our study such as the association between the biofilm formation and the presence of PAIs and antimicrobial resistance as well as the association between the presence of PAIs and phylogeny. Studying these relationships are considered as a significant tools to enhance the knowledge of E. coli populations and the association between strains and disease. 

Minor comments:

  • In Figures 1,2, S1-S3, the interpretations of samples, bands, etc., should be provided. Figure legends should be able to stand alone. Readers should be able to understand them without referring to the text.

           The figure legends were modified as suggested by the reviewer.

  • Lines 212-214:"Chicken and human E. coli were grouped into three major clusters (A, B, C), where isolates in each cluster had identical profiles of PAIs (Fig. 5)". In Fig. 5 appropriate symbols (A, B, C) should be added.

We have addressed this comment by adding the A, B and C symbols to Figure 5.

  • The section “3. Discussion” can be significantly improved by considering these and other recent works:

Gultekin, E. O., Ulger, S. T., & Delialioğlu, N. (2022). Distribution of Pathogenicity Island Markers and Virulence Factors Genes of Extraintestinal Pathogenic Escherichia coli Isolates. Jundishapur Journal of Microbiology, 15(4).

Kuznetsova, M. V., Maslennikova, I. L., Pospelova, J. S., Bertok, D. Ž., & Erjavec, M. S. (2022). Differences in recipient ability of uropathogenic Escherichia coli strains in relation with their pathogenic potential. Infection, Genetics and Evolution, 97, 105160.

El-Shaer, S., Abdel-Rhman, S. H., Barwa, R., & Hassan, R. (2021). Genetic characterization of extended-spectrum β-Lactamase-and carbapenemase-producing Escherichia coli isolated from Egyptian hospitals and environments. PloS one, 16(7), e0255219.

El-Mahdy, R., Mahmoud, R., & Shrief, R. (2021). Characterization of E. coli phylogroups causing catheter-associated urinary tract infection. Infection and Drug Resistance, 14, 3183.

            The suggested reference were cited in the text as suggested by the reviewer.

Reviewer 2 Report

This study aimed to evaluate E. coli isolates from retail chicken meat and human urinary tract infections (for phylogenetic diversity, presence of pathogenicity islands markers, antimicrobial susceptibility phenotypes, and antimicrobial-resistance genes and to evaluate their biofilm formation capacity.

Despite these important findings, the authors must address some areas of concern.

Areas of concern:

Abstract

 This section lacks the background and the methodology. In addition, the biofilm formation potential of the E.coli isolates is missing.

Introduction

Lines 103-104: the following statement is not clear: ‘‘Even though PAIs have been reported extensively in E.coli 103 isolated from UTIs, these PAIs characterization from chicken meat origin rarely concerned 104 before’’

Results

Table 3: could there be any reason explaining the absence of PAI III536, PAI ICFT073, PAI II536, PAI I536, and PAI IIJ196 both in chicken and human isolates? Are they naturally rare PAIs?

Lines 170-171: ‘‘Interestingly, four isolates from chicken (n=1) and human (n=3) were pan-resistant to all antimicrobials used in this study’’. You should have outlined their phylotypic profile, their PAIs distribution, and biofilm formation capacity in trying to understand their pan-resistance.

Table 6: You needed to show the influence of biofilm formation on the AMR potential of E.coli isolates.

Discussion

Lines 267-271: Why are you repeating the results? You are expected to give meaning to your results.

Lines 340-342: The authors failed to demonstrate the impact of biofilm formation on antimicrobial resistance of E.coli isolates in this study

Materials and methods

Line 355: Why did you collect 105 samples from chickens and 67 only from humans?

Lines 356-359: How do you explain the fact that you collected samples in 2019 and obtained approval from the Committee on the Ethics of Animal Experiments in 2021? There is no indication that you received ethical clearance for human subjects from whom you collected the urine samples.

References

Check references 3, 61, 64, and 73 for consistency.

Author Response

First, we would like to thank the reviewer for his time spent carefully reviewing the manuscript. We made sure that each one of the reviewer comments has been addressed carefully and the paper is revised accordingly.

Areas of concern:

Abstract

This section lacks the background and the methodology. In addition, the biofilm formation potential of the E.coli isolates is missing.

We thank the reviewer for pointing this out. We have revised the abstract and all the suggestions were considered.

Introduction

Lines 103-104: the following statement is not clear: ‘‘Even though PAIs have been reported extensively in E.coli 103 isolated from UTIs, these PAIs characterization from chicken meat origin rarely concerned 104 before’’

We have revised the statement to address your concerns and hope that it is now clearer. Please see page 3 in the revised manuscript,  lines 108-110.

Results

Table 3: could there be any reason explaining the absence of PAI III536, PAI ICFT073, PAI II536, PAI I536, and PAI IIJ196 both in chicken and human isolates? Are they naturally rare PAIs?

The high frequency of PAI IV536 was explained by many authors as this PAI is the first to be acquired and fixed on the chromosome, making it the most stable PAI. While the other PAIs like II and III of E. coli 536 were the most unstable, and consequently easy to lose as a result its frequency is commonly (line 282-285)

Lines 170-171: ‘‘Interestingly, four isolates from chicken (n=1) and human (n=3) were pan-resistant to all antimicrobials used in this study’’. You should have outlined their phylotypic profile, their PAIs distribution, and biofilm formation capacity in trying to understand their pan-resistance.

Phylotypic profile, PAIs distribution, and biofilm formation capacity of the pan- resistant strains were mentioned. Please see Line 346-340 in the revised manuscript.

Table 6: You needed to show the influence of biofilm formation on the AMR potential of E.coli isolates.

We agree with the reviewer and accordingly, a correlation analysis was performed to find the association of antimicrobial resistance with biofilm production. Please see page 8, lines 210-213 and the supplementary Figure 4.

Discussion

Lines 267-271: Why are you repeating the results? You are expected to give meaning to your results.

We agree and have removed the repetition of results and provide an explanation for our results as suggested (line 282-285)

Lines 340-342: The authors failed to demonstrate the impact of biofilm formation on antimicrobial resistance of E.coli isolates in this study

Correlation analysis was performed to find the association of antimicrobial resistance with biofilm production please see lines 210-213.

Materials and methods

Line 355: Why did you collect 105 samples from chickens and 67 only from humans?

In this study, urine samples from humans were taken from patients with urinary tracts infections (UTIs) which is somewhat few compared to the availability of retail chicken samples. The overall aim of this study was achieved by comparing both human and chicken isolates for the existence of PAIs and phylogenetic groups, that is not dependent on number of isolates from both sources.

Lines 356-359: How do you explain the fact that you collected samples in 2019 and obtained approval from the Committee on the Ethics of Animal Experiments in 2021? There is no indication that you received ethical clearance for human subjects from whom you collected the urine samples.

Response: We would like to thank the reviewer for the valuable comments. When we started this experiment in 2019, the ethical committee in Faculty of Veterinary Medicine, Mansoura University hadn’t started work yet, and Board of Faculty authorized department councils to provide approvals for research experiments in each department. The ethical committee became then effective starting from September 2019, however experiments started prior this date still have right to get approvals from the committee if the study had animal experiments or involved invasive samples from humans. For this study, we get approval from the ethical committee in 2021, although our study has neither experimental work on animals nor involvement of invasive samples from humans.

So, to avoid confusion regarding this point, we have omitted this sentence (Lines 402-405)

“the study was conducted according to the guidelines of the Declaration of Helsinki and approved by the Committee on the Ethics of Animal Experiments of the Faculty of Veterinary Medicine, Mansoura University (Permit numbers R/60 on 27 February 2021)”

Regarding consent from patients: all human samples enrolled in this study were taken from patients admitted to Specialized Medical Hospital Mansoura, and informed consents were provided by patients prior sampling.

None of the authors has participated to sampling from patients and this sentence has been added to the methodology to read

“Authors were not involved in the sampling process of urine from patients.”

References

Check references 3, 61, 64, and 73 for consistency.

All  the mentioned references have been corrected

Once again, we appreciate the time you took to revise our work and look forward to exceeding your expectations.